Bibliographic revision of Mesacanthion Filipjev, 1927 (Nematoda: Thoracostomopsidae) with description of a new species from Jeju Island, South Korea

Jeong Raehyuk 1
Tchesunov Alexei V. avtchesunov@yandex.ru 2
Lee Wonchoel wlee@hanyang.ac.kr 1
1 Department of Life Science, Hanyang University , Seoul , South Korea
2 Department of Invertebrate Zoology, Faculty of Biology, Moscow State University , Moscow , Russia
Reimer James
Electronic publication date: 2019 Nov 18
Publication date: 2019
Volume: 7
Electronic Location ID: e8023
Received 2019 Jun 20; Accepted 2019 Oct 10
Copyright: ©2019 Jeong et al.
Copyright year: 2019
Copyright holder: Jeong et al.
License: This is an open access article distributed under the terms of the Creative Commons Attribution License, which permits unrestricted use, distribution, reproduction and adaptation in any medium and for any purpose provided that it is properly attributed. For attribution, the original author(s), title, publication source (PeerJ) and either DOI or URL of the article must be cited.
License URL: https://creativecommons.org/licenses/by/4.0/

Keywords: Free-living marine nematodes, Meiofauna, Taxonomy

Funding: National Institute of Biological Resources Ministry of Environment (MOE) of the Republic of Korea NIBR201839201 Korea Research Foundation NRF-2017K2A9A1A06051528 The BK21 Plus Program (Eco-Bio Fusion Research Team) 22A20130012352 Ministry of Education (MOE, South Korea) Russian Fund of Basic Research 18-504-51026 This study was supported by a grant entitled “2018 Graduate Program of Undiscovered Taxa” from the National Institute of Biological Resources (NIBR) funded by the Ministry of Environment (MOE) of the Republic of Korea (NIBR201839201), a grant (NRF-2017K2A9A1A06051528) from the Korea Research Foundation, the BK21 Plus Program (Eco-Bio Fusion Research Team, 22A20130012352) funded by the Ministry of Education (MOE, South Korea), and a grant 18-504-51026 Russian Fund of Basic Research. The funders had no role in study design, data collection and analysis, decision to publish, or preparation of the manuscript.

==============================
A new species of the genus Mesacanthion Filipjev, 1927 was discovered during a survey of natural beaches of Jeju Island in South Korea. The new species Mesacanthion jejuensis sp. nov. shares general morphology of the genus such as the outer labial and cephalic setae being situated at the middle of cephalic capsule, well-developed mandibles with two columns united by a curved bar, and three equally sized and shaped teeth shorter than the mandibles. The new species belongs to a group of Mesacanthion species in which spicules are shorter than two anal body diameters. The new species is most closely related to M. pannosum, first discovered in Puget Sound, Washington, in terms of having enlarged cervical setae flap at the end of cephalic capsule, spicules which are shorter than 2 anal body diameter, both supplementary organ and gubernaculum. It can be distinguished from M. pannosum by its stronger inner labial setae, longer outer labial setae, and difference in the index value of b and c’. Along with the description of Mesacanthion jejuensis sp. nov., the genus Mesacanthion Filipjev, 1927 is bibliographically reviewed and revised. Including the new species, a total of 48 species are described within the genus; 39 which are valid; eight which are considered to be species inquirenda due to misplacement of genus and poor description; one which is considered nomen nudum. An updated diagnosis of the genus is provided along with a compiled tabular key comparing different diagnostic morphological characters of all valid species, as well as a pictorial key consisting of 21 species with spicules shorter than two anal body diameters.

Introduction

Over 50 species of free-living marine nematodes have been reported in South Korea, including those reported on domestic journals (Rho & Min, 2011; Barnes, Kim & Lee, 2012; Hong & Lee, 2014; Kim, Tchesunov & Lee, 2015; Hong, Tchesunov & Lee, 2016; Jeong, Tchesunov & Lee, 2019). The majority of the species found in South Korea belong to the family Draconematidae Filipjev, 1918 and other families reported so far includes Comesomatidae Filipjev, 1918, Desmoscolecidae Shipley, 1896, Enchelidiidae Filipjev, 1918 Cyatholaimidae Filipjev, 1918 and Ironidae De Man, 1876. This is the first record of the genus Mesacanthion, let alone the family Thoracostomopsidae Filipjev, 1927 to be recorded in South Korea.

Family Thoracostomopsidae was first erected by Filipjev (1927) and it is composed of three subfamilies: Thoracostomopsinae (Filipjev, 1927) (2 genera), Trileptiinae (Gerlach & Riemann, 1974) (one genus), and Enoplolaiminae De Coninck, 1965 (19 genera). The three subfamilies can be differentiated by the presence or absence of mandibles (Enoplolaiminae or Trileptiinae respectively), with Thoracostomopsinae uniquely bearing a long and eversible spear (Smol & Coomans, 2006). Now total of 238 species belonging to 22 genera make up the family to date (Bezerra et al., 2019). The genus Mesacanthion (Filipjev, 1927) was first erected as a subgenus of Enoplolaimus (De Man, 1893) with type species Mesacanthion lucifer (Filipjev, 1927; Gerlach & Riemann, 1974) discovered from Barents Sea. Filipjev (1927) specified the characters of the genus Mesacanthion to be three short equal (seldom slightly different) onchia, cephalic setae placed in the middle or anterior to the cephalic capsule with tapered tail with a short dactyli/claviform terminal part. Many of the species currently belonging to the genus Mesacanthion were those transferred from the genus Enoplolaimus when Mesacanthion had been newly erected as a subgenus by Filipjev (1927). Most species (98%) belonging to this genus are recorded from marine environments with exception to one species (Mesacanthion alexandrinus Nicholas, 1993), which was recorded in freshwater environment. Of the valid species, 40% (16) were described from Europe; 20% (eight) from America (four from North and South); 17.5% (seven) from Asia (mainly from western Asia), 15% (six) from Africa, and 7.5% (three) from Australia. The genus Mesacanthion is the second most diverse genus in the family next to Enoplolaimus (De Man, 1893), with 40 valid species recorded to date.

The aim of this study was to review the genus by compiling information such as species distribution, tabular and pictorial key of the genus while determining the validity of existing species. In addition to the revision, Mesacanthion jejuensis sp. nov. is described from Jeju Island, South Korea. An updated diagnosis of the genus is provided with a compiled tabular key consisting of all valid species as well as a pictorial key consisting of 21 species with spicules shorter than two anal body diameters.

Materials and Methods

Sampling and morphological study

A series of sampling took place in June 2018, during a survey of natural beaches of Jeju Island, South Korea (Fig. 1). Two sub-samples of the sediments from the intertidal zone were obtained using a 10 cm2 acryl sampling tube. Sediments were fixed in 5% neutralized formalin solution and brought back to the laboratory. Meiofauna were extracted using the Ludox method (Burgess, 2001), and post-fixed with 70% ethanol dyed with Rose bengal. Nematodes were counted and individual specimens of interest were picked to a Petri dish filled with 10% glycerin. The dish was placed in a drying oven set at 40 °C for a day or two to be completely dehydrated as conferred in the glycerin-ethanol method (Seinhorst, 1959). A single or as many as five specimens (depending on their size) were mounted in a single drop of anhydrous glycerin on a glass slide using the wax-ring method (Hooper, 1986). Mounted specimens were identified under Olympus BX51, Leica DM5000B and DM2500 microscopes. All morphometric measurements were done manually using IC measure v.2.0.0.161 software. For scanning electron microscopy, specimens were placed in a drop of glycerin and gradually mixed with drops of distilled water to be washed from any remnant of glycerin. Hydrated specimen were treated to ethanol series for dehydration (20%, 40%, 50%, 70%, 80%, 90%, 95%, 100%, for 10 min each) and then placed in hexamethyldisilazane (HMDS). Specimens bathed in HMDS were placed in a drying oven to be dried. Once dried, specimens were mounted on a stub to be splutter coated, and observed with COXEM EM-30 microscope.

Figure 1 Map of sampling locality.

This map is made with QGIS software v.2.18.14, a free and open source geographic information system (https://qgis.org).

Revision of the genus

The Bremerhaven Checklist of Aquatic Nematodes by Gerlach & Riemann (1974) was used as primary referral when collecting original descriptions/references and additional information on their distribution. Any updates and changes made to the genus subsequent to 1974 were checked using NeMys, World Database of Nematodes. Once all references had been collected; (1) tabular key consisting of diagnostic characters of all valid species were compiled, (2) distribution of species were determined, (3) validity of each species were determined via comparison and examination, (4) diagnosis of the genus was updated. To construct a pictorial key, original depictions were collected from respective papers and their heads, tails and spicules (if available) were resized and oriented using Adobe Photoshop CS6 for optimum comparison between species. The original drawings were retraced using Wacom Intuous Pro Pen Tablet and Adobe Photoshop CS6.

Nomenclatural acts

The electronic version of this article in Portable Document Format (PDF) will represent a published work according to the International Commission on Zoological Nomenclature (ICZN), and hence the new names contained in the electronic version are effectively published under that Code from the electronic edition alone. This published work and the nomenclatural acts it contains have been registered in ZooBank, the online registration system for the ICZN. The ZooBank LSIDs (Life Science Identifiers) can be resolved and the associated information viewed through any standard web browser by appending the LSID to the prefix http://zoobank.org/. The LSID for this publication is: urn:lsid:zoobank.org:pub: 989DF431-166A-4534-9A37-9AC408194DE7. The online version of this work is archived and available from the following digital repositories: PeerJ, PubMed Central and CLOCKSS.

Systematics

Order Enoplida Filipjev, 1929	
Family Thoracostomopsidae Filipjev, 1927	
Subfamily Enoplolaiminae De Coninck, 1965	
Genus MesacanthionFilipjev, 1927	

Generic diagnosis: (Updated from Wieser, 1953; Platt & Warwick, 1983; Smol, Muthumbi & Sharma, 2014) Enoplolaiminae. Outer labial and cephalic setae situated at middle or anterior end of cephalic capsule. Mandible well-developed, provided with claws, arch-shaped, consisting of two rod-like columns anteriorly united by a curved bar. Teeth shorter than mandibles. Spicule mostly short, unipartite and symmetrical, sometimes long, bipartite (divided by a seam: M. ditlevseni) and asymmetrical (anisomorphic and anisometric: M. diplechma). If long, usually gubernaculum present with caudal apophysis. Marine and freshwater.

Type species: Mesacanthion lucifer (Filipjev, 1927) Gerlach & Riemann, 1974.

Notes on generic diagnosis: Mesacanthion, Enoplolaimus De Man, 1893, Paramesacanthion Wieser, 1953, and Oxyonchus Filipjev, 1927 bear mandibles which are arch-shaped, consisting of two rod-like columns while mandibles of Enoploides are solid, two lateral bars fused to form a single rod. Oxyonchus can be distinguished from other genera which bear similar mandibles by its two uniquely large ventrosublateral teeth which extend to anterior end of the mandibles with small dorsal tooth. Mesacanthion, Paramesacanthion, Enoplolaimus, all have teeth shorter than the mandibles, but the latter can be distinguished by the placement of their outer labial and cephalic setae at posterior end of cephalic capsule. Mesacanthion species have their outer labial and cephalic setae at the middle or anterior end of the cephalic capsule, similar to Paramesacanthion species except outer labial and cephalic setae are only in front of anterior end of cephalic capsule. Mesacanthion and Paramesacanthion share the most characters, making them the closest related genera within the family. The two genera can be differentiated from each other however by the following three characteristics: 1. Outer labial and cephalic setae are located at the anterior end of cephalic capsule for Paramesacanthion while outer labial and cephalic setae are located at the middle or anterior end of cephalic capsule for Mesacanthion. Paramesacanthion species have extra ring(s) of subcephalic setae located at the middle of cephalic capsule where outer labial and cephalic setae would be located for Mesacanthion species. This means when compared to Mesacanthion species, Paramesacanthion species may appear to have extra ring(s) of setae at the anterior end of cephalic capsule, in between inner labial setae and cephalic setae/outer labial setae. This seemingly additional ring of setae are the true cephalic setae, while ring of setae at the middle of cephalic capsule are actually the sub-cephalic setae for Paramesacanthion species; 2. Sexual dimorphism is apparent in the pilosity of the head for Paramesacanthion species, while it is not apparent in Mesacanthion species; 3. All Paramesacanthion species have spicules consisting of two portions, distal and proximal, articulating from one another, while only some Mesacanthion species (M. audax, M. ditlevseni, M. infantile and M. jejuensis sp. nov.) have bipartite spicules divided by a transversal seam, but without the obvious articulation or constriction.

List of valid species

1. Mesacanthion africanthiforme Warwick, 1970 (Warwick, 1970: 142–145, fig. 2A–E; three males and three females, Exe estuary, England).

2. Mesacanthion africanum Gerlach, 1957 (Gerlach, 1957b: 4, fig. 3A–C; description based on one male, Atlantic at Congo mouth, plankton net from above muddy ground).

3. Mesacanthion agubernatus Vitiello, 1971 (Vitiello, 1971: 860, fig. 1A–E; description based on one male, Mediterranean, terrigenous coastal muds, 60 m deep).

4. Mesacanthion alexandrinus Nicholas, 1993 (Nicholas, 1993: 163, 165, fig. 1A–E, 2A–D; four males and three females, sand at water edge of fresh-water Lake Alexandrina, South Australia).

5. Mesacanthion arabium Warwick, 1973 (Warwick, 1973: 114–116, fig. 14A–G; three males and three females, Arabian Sea, fine sand, 49 m deep).

6. Mesacanthion arcuatile Wieser, 1959 (Wieser, 1959: 16–17, Pl. 11 fig. 11A–B; description based on one female, Alki Beach, Washington, US, 6.5 ft, lapsus arcuatilis).

7. Mesacanthion armatum Timm, 1961 (Timm, 1961: 32, fig. 5A–C; more than one male and one female, Bay of Bengal, on Siphonocladus, lapsus armatus).

8. Mesacanthion audax (Ditlevsen, 1918) Filipjev, 1927 [Ditlevsen, 1918: 208–209, pl. 14 fig. 4, 7, pl. 15 fig. 5 (=Enoplolaimus audax); description based on one male, Øresund, off Aalsgaarde. Filipjev, 1927: 143; transfer Enoplolaimus audax to subgenus Mesacanthion. Gerlach, 1958b: 73; (as Mesacanthion audax), Kiel Bay, Sand and silt, 6 m deep. Riemann, 1966: 186; three males, North Sea, sand].

9. Mesacanthion banale (Filipjev, 1927) Gerlach & Riemann, 1974 [Filipjev, 1927: 147, Pl. 7 fig. 40A, B; (=Enoplolaimus (Mesacanthion) banalis), description based on three females, Barents Sea, muddy sand, 25 m deep. Gerlach & Riemann, 1974: 531; transfer Enoplolaimus banale to genus Mesacanthion].

10. Mesacanthion breviseta (Filipjev, 1927) Gerlach & Riemann, 1974 [Filipjev, 1927: 150–151, pl. 7 fig. 43A–C; (=Enoplolaimus (Mesacanthion) breviseta) description based on one male and a juvenile male, Barents Sea, sand with shells and stones, 83 m deep. Gerlach & Riemann, 1974: 531; transfer Enoplolaimus (Mesacanthion) breviseta to genus Mesacanthion].

11. Mesacanthion cavei Inglis, 1964 [Inglis, 1964: 313–314, fig. 76–78; description based on two males (one in poor condition) and one damaged juvenile, South Africa, coarse sand and broken shells, 26–27 m deep].

12. Mesacanthion ceeum Inglis, 1964 (Inglis, 1964: 313, fig. 74–75; description based on one male and one juvenile, South Africa, coarse sand and broken shells, 26 m deep, lapsus ceeus).

13. Mesacanthion conicum (Filipjev, 1918) Filipjev, 1927 [Filipjev, 1918: 105–107, Table 3, fig. 16A–B; (=Enoplolaimus conicus), description based on one female, Black Sea. Filipjev, 1927: 143; transfer Enoplolaimus conicus to subgenus Mesacanthion].

14. Mesacanthion cricetoides Wieser, 1959 (Wieser, 1959: 17–18, fig. 13A–B; description based on one female, Richmond Beach, Washington, 2.5 ft deep).

15. Mesacanthion diplechma (Southern, 1914) Filipjev, 1927 [Southern, 1914: 55–56, fig. 25A–J; (=Enoplus diplechma), two males and two females, Clew Bay, sandy bottom, 25–31 m deep. Filipjev, 1927: 143; transfer Enoplus diplechma to subgenus Mesacanthion. Gerlach, 1958: 72; as Mesacanthion diplechna, Kiel Bay, silt, 8 m deep. Riemann, 1966: 186 North Sea, sand. Boucher, 1977: 741–743, Figs. 4A–4E; as Mesacanthion diplechma Southern, 1914, one male, three females and six juveniles, Pierre Noire (Western Channel), infralittoral sands].

16. Mesacanthion ditlevseni (Filipjev, 1927) Gerlach & Riemann, 1974 [Filipjev, 1927: 148, pl. 5 fig. 41A–D; (=Enoplolaimus (Mesacanthion) ditlevseni), three males and one female, Barents Sea, silt with stones, 36–280 m deep. Ditlevsen, 1928: 210–213, fig. 8–13; (=Enoplolaimus angustignathus), one male and one female, Greenland, mud, clay, 100–200 m deep, De Coninck and Stekhoven, 1933: 38. Allgén, 1954: 22; (as Enoplolaimus (Mesacanthion) angustignathus), five males and nineteen females, Jan Mayen, Greenland, black sand, 23 m deep. Gerlach & Riemann, 1974: 532; transfer Enoplolaimus ditlevseni to genus Mesacanthion].

17. Mesacanthion fricum Inglis, 1966 (Inglis, 1966: 87, fig. 10–12; description based on one male, South Africa, sand, lapsus frica).

18. Mesacanthion heterospiculum Sergeeva, 1974 (Sergeeva, 1974: 123, fig. 4A–4B; description based on 14 males, Black Sea, various depths and sediments).

19. Mesacanthion hirsutum Gerlach, 1953 (Gerlach, 1953: 536–537, fig. 9A–E; description based on one male and one female, Mediterranean. Gerlach, 1967: 26, fig. 10A–E; two males, two juveniles and one male, Sarso Island, Red Sea, Saudi Arabia).

20. Mesacanthion infantile (Ditlevsen, 1930) De Coninck & Schuurmans Stekhoven, 1933 [Ditlevsen, 1930: 205–208, fig. 8–10; (=Enoplolaimus infantilis), one male and one female, Stewart Island, Halfmoon Bay, sand, 5–7 fms. Allgén, 1951: 322–323, fig. 33A–B; (=Enoplolaimus mortenseni), description based on one female, Australia see Mawson, 1956: 65–66 (re-examination of type specimen=Mesacanthion infantilis), op Wieser, 1953: 75. Allgén, 1951: 323–324, fig. 34A–B; (=Enoplolaimus philippinensis), description based on one juvenile, Australia, op Mawson, 1956: 65–66 (re-examination of type specimen=Mesacanthion infantilis). De Coninck & Schuurmans Stekhoven, 1933: 38; (as Mesacanthion infantile). Wieser, 1953: 76, fig. 39A–Btwo females, Chile. Mawson, 1956: 65–66, fig. 29A–C; two juveniles, Antarctica].

21. Mesacanthion karense (Filipjev, 1927) Gerlach & Riemann1974 [Filipjev, 1927: 152, pl. 7 fig. 45A–C; (=Enoplolaimus (Mesacanthion) karensis), one juvenile male and three females, Kara Sea, sand, 15 m deep. Gerlach & Riemann, 1974: 533; transfer Enoplolaimus (Mesacanthion) karensis to genus Mesacanthion].

22. Mesacanthion kerguelense Mawson, 1958 (Mawson, 1958: 338–339, fig. 22A–D; five males, two females and three juveniles, Kerguelen Island, Heard Island, Macquarie Island).

23. Mesacanthion longispiculum Gerlach, 1954 (Gerlach, 1954: 228–229, fig. 1A–B; one male and one female, Mediterranean. Gerlach, 1957a: 421; Brazil. Gerlach, 1958a: 352–353, fig. 4A–C; (as cf. longispiculum), one male, Mananjary, Madagascar, muddy sand).

24. Mesacanthion longissimesetosum Wieser, 1953 (Wieser, 1953: 78–79, fig. 42A–E; two males, one female and thirteen juveniles, Chile, littoral exposed and sheltered sand, sublittoral secondary substratum and soft bottom, lapsus longissimesetosus).

25. Mesacanthion lucifer (Filipjev, 1927) Gerlach & Riemann, 1974 [Filipjev, 1927: 149–150, pl. 7 fig. 42A–C; (=Enoplolaimus (Mesacanthion) lucifer), one male and two females, Barents Sea, Kara Sea, sand and sandy silt, 18–83 m deep. (Gerlach & Riemann, 1974): 533; transfer Enoplolaimus (Mesacanthion) lucifer to genus Mesacanthion.]

26. Mesacanthion majus (Filipjev, 1927) Gerlach & Riemann, 1974 [Filipjev, 1927: 151–152, pl. 7 fig. 44A–C; (=Enoplolaimus (Mesacanthion) major), three females, Kara Sea, Barents Sea, sand and gravel, 15–36 m deep. Wieser, 1953: 78, fig. 41A–D; (as Mesacanthion major (Filipjev, 1925b), four males, two females and 15 juveniles, Arctic Sea, Chile, sublittoral, secondary substratum and coarse bottom, lapsus major. Gerlach & Riemann, 1974: 533; (as Mesacanthion majus Filipjev, 1927).]

27. Mesacanthion marisalbi Galtsova, 1976 (Galtsova, 1976: 261–263, fig. 7; two males, one female and one juvenile, White Sea, littoral zone in slightly silted sand).

28. Mesacanthion monhystera Gerlach, 1967 (Gerlach, 1967: 27–28, fig. 11A–F; one male and one juvenile female, Red Sea, sandy beach and littoral subsoil water).

29. Mesacanthion obscurum Gagarin & Klerman, 2006 (Gagarin & Klerman, 2006: 533–535, fig. 1A–E; twelve males and eight females, Mediterranean Sea off the Israeli coast near Hadera, sandy sediment, 30–35 m deep).

30. Mesacanthion pali Wieser1959 (Wieser, 1959: 16, fig. 10A–B; description based on one male, Puget Sound, subterranean water, medium fine to coarse sand).

31. Mesacanthion pannosum Wieser, 1959 (Wieser, 1959: 17, fig. 12A–D; one female and one female, Puget Sound, medium fine to coarse sand, 2.5 ft deep).

32. Mesacanthion propinquum Gagarin & Klerman, 2006 (Gagarin & Klerman, 2006: 536–538, fig. 2A–E; twelve males and eleven females, Mediterranean Sea off the Israeli coast near Hadera, sandy sediment, 30–35 m deep).

33. Mesacanthion proximum Gerlach, 1957 (Gerlach, 1957a: 427–429, fig. 5G–5M; one male and one juvenile, Santos, Brazil, fine sand).

34. Mesacanthion rigens Gerlach, 1957 (Gerlach, 1957a: 427, fig. 5C–5F; one male and one female, Bertioga, Brazil. Gerlach, 1956: 204; Brazil, nomen nudum).

35. Mesacanthion southerni Warwick, 1973 (Warwick, 1973: 111–114, fig. 12A–C, 13A–C; six males, three females and two juveniles, Arabian Sea, fine sand and fine muddy sand, 48–49 m deep).

36. Mesacanthion studiosum Inglis, 1964 (Inglis, 1964: 315–316, fig. 79–90; two males, two females and two juveniles, South Africa, coarse white sand, 27 m deep, lapsus studiosa).

37. Mesacanthion tenuicaudatum (Ssaweljev, 1912) De Coninck & Schuurmans Stekhoven, 1933 [Ssaweljev, 1912: 111–112; (=Enoplolaimus tenuicaudatus), both sex but number of specimen not specified, White Sea, lapsus tenuicaudatus. De Coninck & Schuurmans Stekhoven, 1933: 39; transfer and correct name from Enoplolaimus tenuicaudatus to Mesacanthion tenuicaudatum].

38. Mesacanthion virile (Ditlevsen, 1930) De Coninck & Schuurmans Stekhoven, 1933 [Ditlevsen, 1930: 208–211, fig. 11–14; (=Enoplolaimus virilis), description based on one male, Stewart Island; Halfmoon Bay, New Zealand, Sand, 9.1–12.8 m(converted from fathom). De Coninck & Schuurmans Stekhoven, 1933: 39; transfer and correct name Enoplolaimus virilis to Mesacanthion virile. Allgén, 1959: 48–50; 8 females and twelve juveniles, Falkland Islands, South Georgia, Graham Land].

Species Inquirenda

1. Mesacanthion brachycolle Allgen, 1959 [Allgén, 1959: 50, fig. 32A, B; two females and two juveniles, Falkland Islands, sandy bottom with algae, 40 m deep, Graham Island, mud, 125 m deep. Allgén, 1960: 479, fig. 3; (as Enoplolaimus (Mesacanthion) brachycollis), lapsus brachycollis, one female and one juvenile, Falkland Islands]. Species Inquirenda. This species is placed as species inquirenda due to the following reasons: (1) substandard quality of the original text and figures making it impossible to understand, to which genus this species should be referred as; (2) ambiguity of the material, where females and juveniles are indicated as the only materials yet a male tail is given on fig. 32B.

2. Mesacanthion donsitarvae (Allgen, 1935) Wieser, 1953 (species inquirenda) [Allgén, 1935: 47; (=Enoplolaimus donsitarvae) Norway, lapsus (donsi)-tarvae. Wieser, 1953: 76; transfer Enoplolaimus donsitarvae to genus Mesacanthion and opinionates the fact that Allgén provided no figures and description was based on erroneous data of Ditlevsen on wrong number of cephalic setae].

3. Mesacanthion gracilisetosum (Allgen, 1930) Wieser, 1953 (species inquirenda) [Allgén, 1930: 189–191, Figs. 1–3; (=Enoplolaimus gracilisetosus), one male, two females and one juvenile, Macquarie Island. Wieser, 1953: 76; transfer Enoplolaimus gracilisetosus to genus Mesacanthion, lapsus gracilisetosus ].

4. Mesacanthion hawaiiense (Allgen, 1951) Wieser, 1953 (species inquirenda) [Allgén, 1951: 274–275, fig. 5A–5B; (=Enoplolaimus hawaiiensis), description based on one female, Honolulu, Hawaii. Wieser, 1953: 75; transfer Enoplolaimus hawaiiensis to genus Mesacanthion and opinionates description is insufficient, lapsus hawaiiensis ].

5. Mesacanthion pacificum (Allgen, 1947) Wieser, 1953 (species inquirenda) [Allgén, 1947: 212, fig. 76A–B; (=Enoplolaimus pacificus), description based on one female and one juvenile, Bay of Panama, Perlas Island. Allgén, 1951: 275, 277, Figs. 6A–6D; one male, one female and three juveniles, Coast of Honolulu. Wieser, 1953: 66, 76; transfer Enoplolaimus pacificus to genus Mesacanthion and opinionates it resembles Oxyonchus more. Allgén, 1959: 48; (as Mesacanthion pacificus), two juveniles, Falkland Islands, sand and small stones with algae, 40 m deep, lapsus pacificus].

6. Mesacanthion paradentatum (Allgen, 1932) Wieser, 1953 (species inquirenda) [Allgén, 1932: 111–112, fig. 8A–B; (=Enoplolaimus paradentatus), description based on one juvenile, Campbell Island. Wieser, 1953: 76; transfer Enoplolaimus paradentatus to genus Mesacanthion, lapsus paradentatus].

7. Mesacanthion primitivum (Allgen, 1929) Wieser, 1953 (species inquirenda) [Allgén, 1929: 441, fig. 6A–B; (=Enoplolaimus primitivus), Skagerrak. Wieser, 1953: 76; transfer Enoplolaimus primitivus to genus Mesacanthion, lapsus primitivus].

8. Mesacanthion ungulatum (Wieser, 1953) Wieser, 1953: 78, fig. 40A–B; description based on two juveniles, Seno Reloncavi proper, Chile, exposed littoral algae, lapsus ungulatus). Species inquirenda. Further discussed in the discussion.

Nomen nudum

1. Mesacanthion microsetosus Allgen, 1932 (nomen nudum –Bezerra et al., 2019) [Allgén, 1932: 110–111, fig. 7A–B; (=Enoplolaimus microsetosus) description based on one juvenile, Campbell Island 40 m deep. Allgén, 1959: 48; (transfer Enoplolaimus microsetosus to genus Mesacanthion) nine females and five juveniles, South Georgia, Antarctica, clay with sparse stones, 125 m deep, lapsus microsetosus, nomen nudum]. Only female or juvenile used for description and according to Wieser, 1953, Allgén stating four labial and four cephalic setae makes his description doubtful, Wieser, 1953: 82; moved to Paramesacanthion.

Mesacanthion jejuensis sp. nov.	
Figs. 2 and 3, Table 1	
urn:lsid:zoobank.org:act:EE4EB2FC-59DA-48D3-9C10-C9E5646AF0D9	

Type locality: Intertidal zone at coast of Jeju Island, South Korea (33°26′05″N 126°55′15″E), in sandy beach

Type material: All specimen deposited in National Institute of Biological Resources (South Korea). Holotype 1♂ (NIBRIV00008488276) on one slide, Allotype 1♀ (NIBRIV00008488277) on one slide, Paratypes 2♂♂, 1♀ on two different slides (NIBRIV00008488278–NIBRIV00008488279), 1♂ and 1♀ dried, mounted on two separate stubs and coated with gold for SEM (NIBRIV00008488280–NIBRIV00008488281) from coast of Jeju Island, South Korea (33°26′05″N 126°55′15″E) collected on 17 June 2018.

Measurements: See Table 1 for detailed measurements and morphometric ratios.

Figure 2 Mesacanthion jejuensis. sp. nov. male.

(A) Head, lateral view. (B) Tail, with spicule and gubernaculum. (C) Total view. (D) Bipartite spicules with triangular gubernaculum. Scale bars: 20 μm (A, B, and D) and 200 μm (C). Figure credit: Raehyuk Jeong.

Figure 3 Mesacanthion jejuensis. sp. nov. female.

(A) Head, lateral view. (B) Reproductive system with vulva protruding. (C) Total view. (D) tail region with caudal glands. (E) Ventrosublateral mandible. Scale bars: 20 μm (A and D),100 μm (B and C) and 10 μm (E). Figure credit: Raehyuk Jeong.

Description: Male (Fig. 2). Cuticle smooth above cephalic capsule, finely striated posterior to cephalic capsule until tail tip. Three lips well developed; edges of lips narrowed and distally pointy curving outwards, each lips carrying two inner labial setae. Six inner labial setae, stout and conical 12 µm long. Six longer outer labial setae and four shorter cephalic setae sharing one crown, situated at midlevel of cephalic capsule. Cephalic capsule vaguely set off at mid-level, anterior part narrow, and posterior part gradually thicker. Buccal cavity funnel shaped, wide at anterior end, gradually narrowing to the base. Coffee-bean shaped epidermal glands distributed along dorsal plane from anterior end of body until posterior end. Buccal cavity armed with three mandibles and three teeth. Mandible consisting of two rods distancing from one another anteriorly joined by anterior rod. Lateral edges of each rod with teeth or denticle pointing to the lumen (Fig. 3E). ∼5–6 short cervical setae in singles at level of posterior end of cephalic capsule. Modified cervical setae, a flap, inverse triangular, just posterior to a single lateral outer labial setae at posterior end of cephalic capsule, observable in all four males on both lateral body sides (Fig. 4B). Amphid ambiguously present below the cervical flap, pouch-shaped. Two pores observed diagonally below cervical flap and amphid (Fig. 4B). Cervical somatic setae in 8 groups of 2–3 around pharyngeal region a, roughly two cephalic capsule lengths below level of cephalic capsule end (Figs. 4A and 4B). Some cervical setae partly possessing irregular lateral and terminal processes, resembling penicillus or plumule (Figs. 4A and 4B). Somatic setae scarcely distributed along the body in singles until tail region. Pharynx fairly long and annulated with plasmatic lens-like interlayers and sinuous external contours, cardia triangular and going into the middle of intestine. Metanemes not visible. Testes paired opposed, both ends situated to the right of the intestine. Thick supplement, 18 µm long, 165 µm above from cloacal opening. Spicules paired, bipartite, symmetrical, curved slightly and thick. Each spicule with distinct transverse, oblique seam, dividing it distal and proximal portions (Figs. 5A and 5B). Distal portion shorter than proximal portion. Distal portion slightly curved towards cloacal opening, anterior end with one denticle just above and/away from its round pointy end. Proximal portion rather straight, posterior end with a knob/neck-like constriction. Gubernaculum embracing spicules, shaped like irregular triangle, lateral end which lies lateral to the spicule, almost perpendicular to axis of the anus, even extending beyond distal end of spicule, and the other end arching off at an angle towards the tail. Tail elongated and papilliform. five somatic setae in tail region. Caudal gland protruded anterior to the anus, their nucleus-containing bodies located along the posterior midgut. Spinneret well developed. 1 short caudal (terminal) setae (with porous) just above distal end of tail.

Table 1 Measurement of diagnostic morphological characters of Mesacanthion jejuensis sp. nov.

Measurements are in μm where applicable, and morphometric values rounded.

Characters	♂ holotype	♂ (n = 4) mean ± sd (range)	♀ (n = 3) mean ± sd (range)	
Body length	3682	3401 ± 476 (2703–3723)	3719 ± 808 (3080–4627)	
Maximum body diameter	79	79 ± 3 (76–82)	108 ± 31 (80–141)	
Diameter at the level of cephalic setae	39	36 ± 3 (32–39)	38 ± 7 (34–46)	
Length of inner labial setae	12	13 ± 2 (11–15)	12 ± 1 (11–13)	
Length of outer labial setae	43	51 ± 7 (43–59)	41 ± 5 (38–47)	
Length of cephalic setae	28	28 ± 7 (18–34)	25 ± 1 (24–26)	
Distance from anterior to cephalic setae	19	15 ± 4 (11–19)	16 ± 4 (13–21)	
Width at cephalic capsule end	42	43 ± 2 (41–45)	45 ± 7 (40–53)	
Length of cephalic capsule	29	28 ± 2 (25–30)	30 ± 5 (26–36)	
Buccal cavity length	50	44 ± 5 (37–50)	44 ± 6 (38–49)	
Distance from nerve ring from anterior end	212	202 ± 28 (161–220)	204 ± 9 (194–209)	
Pharynx (oesophagus) length	731	706 ± 74 (598–764)	706 ± 16 (687–715)	
Corresponding body diameter at pharynx	76	76 ± 2 (74–78)	97 ± 23 (76–122)	
Cardia length	21	23 ± 2 (21–25)	23 ± 4 (18–26)	
Tail length	287	275 ± 44 (209–304)	286 ± 48 (257–342)	
Anal body diameter	50	54 ± 4 (50–60)	58 ± 12 (48–71)	
c’	5.7	5.1 ± 0.8 (4–5.7)	4.9 ± 0.4 (4.6–5.4)	
Length of conical tail	223.0	209 ± 35 (157–229)	227 ± 40 (203–273)	
Length of cylindrical tail	64	66 ± 11 (52–78)	59 ± 9 (52–69)	
Cylindrical tail length portion as percentage of tail length	0	0.3 ± 0 (0.3–0.3)	0.3 ± 0 (0.3–0.3)	
Spicule length as arc	76	79 ± 6 (72–85)	n/a	
Spicule length as arc / anal body diameter	1.5	1.5 ± 0.1 (1.4–1.6)	n/a	
Length of gubernaculum	50	45 ± 5 (39–50)	n/a	
Supplementary organ length	18	15 ± 3 (10–18)	n/a	
Distance from cloacal opening to supplementary organ	165	160 ± 16 (136–171)	n/a	
Distance from anterior end to vulva	n/a	n/a	2027 ± 459 (1685–2549)	
Corresponding body diameter at vulva	n/a	n/a	108 ± 31 (80–141)	
Distance from anterior end to vulva as percentage of total body length	n/a	n/a	54 ± 1 (54–55)	
a	46.6	43.1 ± 5.1 (35.6–46.6)	34.9 ± 3.1 (32.8–38.5)	
b	5	4.8 ± 0.2 (4.5–5)	4.6 ± 0.2 (4.5–4.8)	
c	12.8	12.4 ± 0.6 (11.7–12.9)	12.9 ± 0.8 (12–13.5)	

Figure 4 Scanning electron micrograph of Mesacanthion jejuensis sp. nov.

(A) Male, head region, lateral view, groups of cervical setae in doubles/trios with irregular lateral and terminal processes. (B) Male, head region showing contour of cephalic capsule end and triangular cervical setae flap just posterior to lateral outer labial seta. (C) Male, cloacal opening with distal end of gubernaculum peeking out. (D) Female, head region, lateral view, single cervical seta.

Figure 5 Mesacanthion jejuensis sp. nov. (A and B, paratype).

(A) Lateral view of male cloacal region, showing a seam separating spicules in distal and proximal portions. (B) Lateral view of male cloacal region, showing distal end of spicule and triangular gubernaculum. Scale bars: 30 μm (A and B).

Female (Fig. 3). Female generally longer and larger in size. Three lips higher in female, edges of lips noticeably stronger in female, distal end aggressively curved, each lips carrying two inner labial setae. No subtle sexual dimorphism found in setae in the head region, other than shorter length outer labial and cephalic setae compared to male. Short knobs on each anterior end of mandible. Female lacking cervical setae flap on cephalic capsule end. Amphid not observed. Groups of cervical setae found in esophageal region in males are in singles as opposed to doubles/trios (Fig. 5D). Vulva located at 55% of total body length with protruding lips. Reproductive system didelphic amphidelphic, both ends flexed inwards. Both ovaries positioned left of the intestine, antidromously reflexed. Tail conico-cylindrical, three somatic setae in tail region with no apparent caudal setae.

Diagnosis: Mesacanthion. Body length 2700–4630 µm. Cuticle finely striated along the body, smooth only in cephalic capsule region, head set off with cephalic capsule. Metanemes not visible. Six inner labial setae 8–15 µm. Six longer outer labial setae 36–59 µm, four shorter cephalic setae 18–34 µm long sharing one crown. Buccal cavity armed with mandible and three teeth. Mandible consisting of two rods distancing from one another anteriorly joined by anterior rod. Lateral edges of each rod with teeth or denticle pointing to the lumen. Buccal cavity 37–61 µm long. 8–9 groups of cervical setae in groups of two to three at stoma region. Cervical setae in single groups in females. Males with testis paired and opposed. Spicule paired, symmetrical, slightly curved, divided into two portions by a seam. Distal portion shorter than proximal. Proximal portion with knob/neck-like end. Gubernaculum paired, shaped like an irregular triangle with caudal apophysis, distal end extending beyond spicules and ventrally towards cloacal opening. Precloacal supplementary organ present. Three to four somatic setae distributed along the tale. Tail conico-cylindrical, c’ 4–5.7, cylindrical portion of the tail constituting about 30% of the entire tail length.

Differential diagnosis: Total of 23 species of Mesacanthion with spicules shorter than 2 abd were examined. Species such as M. arcuatile, M. conicum and M. cricetoides were omitted from examination due to the fact that only female was ever described. Also, species with asymmetrical spicules (anisomorphic and anisometric) were omitted even if the shorter spicule is shorter than 2 abd, since Gagarin & Klerman (2006) already provided a key for those group of species. M. tenuicaudatum which most likely does have spicules shorter than 2 abd, is also omitted from examination and pictorial key as there are no depiction of the specimen available. Description also lacks information regarding gubernaculum, measurements of anal body diameter and length of all setae, making it not feasible for comparison. Lastly, M. virile is included for analysis, despite its lack of information on abd. Although it cannot be confirmed that it possesses spicules which are shorter than 2 abd, given the length of spicule and other relative body measurements, it is likely that this species belongs to this group.

The new species is most similar to M. pannosum as they both share striking resemblance in overall morphology. They both have spicules shorter than 2 abd with presence of both supplementary organ and gubernaculum. Index value of both species (a and c) are within range to each other as well. Most interesting character shared by the two species is the presence of modified/transformed cervical setae seen as a flap. This flap-like appendage is seen as inverse triangle just below a single outer labial seta (lateral seta), at the end of cephalic capsule. It was first observed by Wieser (1959) in males of M. pannosum and it has been a character unique to the species until it now. Like M. Pannosum, the flap is also observed only in males of the new species, and the morphology is in line with those previously seen in M. pannosum. The only difference concerning the flap between the two species is that M. pannosum had two flaps next to each other, but only a single flap is seen in the new species (Fig. 4B).

The new species differs from P. pannosum by having stronger (stout) inner labial setae; longer outer labial setae (43–59 µm vs. 24–25 µm, respectively); difference in the index value of b and c’ (4.5–5 vs. 6.3 and 4.0–5.7 vs. 3, respectively); type of spicules (unipartite vs. bipartite with seam, respective).

The validity of diagnostic value in presence or absence of a seam (bipart spicule), can be questionable considering it is an ambiguous character and could be mistaken from a diffraction caused by a large gubernaculum. Due to this reason, the seam was not given a significant importance in the diagnosis of species within the genus, but the character and species which bear it are still discussed for reference.

Total of four species within the genus Mesacanthion have paired bipartite spicules: M. audax, M. ditlevseni, M. infantile, and M. jejuensis sp. nov. They all have spicules which are shorter than 2 abd, but M. audax is most easily distinguished from the other three species by its lack of supplementary organ. While the three species have supplementary organ, the new species resembles M. ditlevseni the most. They both have stout inner labial setae with presence of both supplementary organ and triangular gubernaculum. Their index value (a, b, and c’) are also within range from one another. The new species can be distinguished by its proportionally longer cephalic setae (in which is double the length of cephalic setae observed in M. ditlevseni); presence of cervical flap in new species; dense distribution of cervical setae in groups of doubles/trios below at stoma region in male; differing details to its spicules and gubernaculum. The spicule of the new species is different from M. ditlevseni in that distal portion of the spicule is shorter than proximal portion where vice versa is true for M. ditlevseni. This is peculiar characteristic unique to the new species, as all bipartite spicules found in Mesacanthion species have longer distal portion over proximal portion. We believe that diagnostic value of modified cervical setae flap is greater than bipartite spicule (spicule with seam) in terms of ambiguity. Therefore, the species most similar to the new species is considered to be M. pannosum.

Etymology: The species name jejuensis is given as the species was discovered from coast Jeju Island, South Korea.

Pictorial key to species with spicules shorter than 2 anal body diameter within the genus Mesacanthion and morphometric values for valid species of Mesacanthion Figs. 6–8, Table 2

Key to species with spicules shorter than 2 anal body diameters within the genus Mesacanthion

1.	Supplementary organ present in males …5	
	-Supplementary organ absent in males …2	
2.	Stout post-cloacal setae present …M. africanthiforme	
	-Stout post-cloacal setae absent …3	
3.	Tail conico-cylindrical …M. karense	
	-Tail conical …4	
4.	Tail long (c =14–19) with typical stoma …M. audax	
	-Tail stout (c= 43–51) with well-developed posterior rods of the stoma … M. armatum	
5.	Gubernaculum present in males …11	
	-Gubernaculum absent in males …6	
6.	Body length over 3000 µm …7	
	-Body length below 2000 µm …8	
7.	Several cervical setae at cephalic capsule with spicules arrow shape with sharp distal end …M. agubernatus	
	-No cervical setae at cephalic capsule with simple spicules slightly curved with blunt distal end …M. studiosum	
8.	Mandible decorated by longitudinal marks …M. rigens	
	-Mandible with no decoration …9	
9.	Somatic setae absent …M. proximum	
	-Somatic setae present …10	
10.	Dorsal tooth missing …M. monhystera	
	-Dorsal tooth present …M. hirsutum	
11.	Enlarged cervical setae (flaps) present below lateral outer labial seta at posterior end of cephalic capsule …12	
	-Enlarged cervical setae (flaps) absent …13	
12.	Inner labial setae stout. Index c’ 4.0–5.7 …M. jejuensissp. nov.	
	-Inner labial setae thin. Index c’ 3 …M. pannosum	
13.	Subventral precloacal setae absent …14	
	-Subventral precloacal setae present …18	
14.	Gubernaculum triangular …15	
	-Gubernaculum simple with dorsal appendage …M. marisalbi	
15.	Gubernaculum with two parts; membranous part and rod-like part which supports it caudally with barbed tip …M. virile	
	-Gubernaculum with just one part …16	
16.	Spicule bipartite …M. ditlevseni	
	-Spicule unipartite …17	
17.	Subcephalic setae absent, supplementary organ 2.2 abd away from anus … M. breviseta	
	-Four subcephalic setae very thin near end of cephalic capsule, supplementary organ 3 abd away from anus …M. lucifer	
18.	Proximal end of spicule with massive process …M. fricum	
	-Proximal end of spicule without massive process …19	
19.	Distal end of spicule with backwardly pointing spines …M. audax	
	-Distal end of spicule with no backwardly pointing spines …20	
20.	Spicule distally dilated …M. majus	
	-Spicule with no dilation …21	
21.	Supplementary organ located close (∼1 spicule length) to proximal end of the spicule …M. kerguelense	
	-Supplementary organ located further (>1.1 spicule length) away from the anus …22	
22.	Inner labial setae stout 12 µm long, Index a ∼30, Index b ∼5, Index c’ 4.2–4.5 …M. longissimesetosum	
	-Inner labial setae stout 24 µm long, Index a ∼50, Index b ∼3, Index c’ 2.8 … M. pali	

Discussion

The genus Mesacanthion currently consist of 39 valid species. Of these valid species, Mesacanthion ungulatum (Wieser, 1953) is most ambiguous in terms of validity. The species was erected from a description based on only two juvenile specimens, with reasoning that the species in question bears extremely long labial setae and high lips. While there is no problem with the validity according to the International Code of Zoological Nomenclature, since a new species can be described at any life stage, it is ambiguous nonetheless in its current state as its distinguishing characters for the species is no longer unique to the species. Labial setae in M. ungulatum is noted to be extremely long, measuring to be 15 µm might have been lengthy for the genus at the time of original description, but currently compared to other species within the genus, it is quite average in length. Mesacanthion arabium (Warwick, 1973) bears labial setae measuring from 23–25 µm, albeit its overall longer body length. Even when comparing proportionally to body length, Mesacanthion fricum (Inglis, 1966) (body length 1650 µm/length of inner labial setae 13 µm) and Mesacanthion hirsutum (Gerlach, 1953) (1155–1982 µm/8–14 µm) have longer proportioned inner labial setae than M. ungulatum (2250–2430 µm/15 µm). The only characteristic discerning this species from the rest is then by its high lips, but even that is questionable, considering later described species such as Mesacanthion alexandrinus (Nicholas, 1993), while it does not mention in the description, depict just as high lips as shown in M. ungulatum. It would be appropriate to consider this a synonymization case, and find a species described after M. ungulatum which is most similar to follow the Principle of Priority. Unfortunately, remaining characteristics of M. ungulatum is very generic to the genus and the fact that there is no male to compare its spicules, gubernaculum and supplementary organ which can be unique to each species, no further action can be taken other than to place it as species inquirendum. There is however 18S ribosomal RNA gene, partial sequence of M. ungulatum available on GenBank. It seems that the sample specimen was obtained from Chile, the type locality, so there is high probability that more information can be gathered regarding adult stage of this species. Hopefully someone can review the species with a sound adult specimen to clear this species of its current status.

Figure 6 Pictorial key to species with spicules shorter than 2 anal body diameters within the genus Mesacanthion.

A–D, species without supplementary organ with gubernaculum; E–J, species with supplementary organ without gubernaculum. Species with bipartite spicules marked with asterisk. Figure source: (A) Warwick (1970). (B) Timm (1961). (C) Ditlevsen (1930). (D) Filipjev (1927). (E) Vitiello (1971). (F) Gerlach (1967). (G) Gerlach (1967). (H) Gerlach (1957a). (I) Gerlach (1957a). (J) Inglis (1964).

Figure 7 Pictorial key to species with spicules shorter than two anal body diameters within the genus Mesacanthion.

A–K, species with supplementary organ and gubernaculum. Species with bipartite spicules marked with asterisk. Figure source: (A) Ditlevsen (1918). (B) Filipjev (1927). (C) Filipjev (1927). (D) Inglis (1966). (E) Mawson (1958). (F) Wieser (1953). (G) Filipjev (1927). (H) Wieser (1953). (I) Platonova & Galtsova (1976). (J) Wieser (1959). (K) Ditlevsen (1930).

Figure 8 Pictorial key to species with spicules shorter than two anal body diameters within the genus Mesacanthion.

A–B, species with supplementary organ and gubernaculum along with triangular cervical setae flaps. Species with bipartite spicules marked with asterisk. Figure source: (A) Wieser (1959). Figure credit: (B) Raehyuk Jeong.

Diagnosis of the genus Mesacanthion provided by Smol, Muthumbi & Sharma (2014), was updated in this study based on our review and findings. The original diagnosis specifically states that spicules are generally short, but if long, gubernaculum with caudal apophysis is to be present. This is true in most cases, but with exceptions in species such as M. brevista, which has one of the longest spicules in the genus (165 µm) with no gubernaculum at all, and M. arabium, which has a pair of short spicules (24 µm) bearing a triangular gubernaculum which resembles a caudal apophysis. In addition, accounts for different types of spicules were added (symmetric/asymmetric, bipartite, etc.), so that later encounters of new species with bipartite spicules is not simple mindedly mistaken as Paramesacanthion as opposed to Mesacanthion, as was the case with us.

The morphology of the spicule is especially diverse in the genus Mesacanthion. Spicules which come in a pair can be either short or long/symmetrical or asymmetrical/straight, L-shaped or arcuate in shape. Spicules, if long can be anisomorphic and/or anisometric and can be bipartite or in whole. The part of it being bipartite can be perplexing when it comes to species identification, as it can lead to wrongful placement of the species to the related genus Paramesacanthion. Diagnosis of Paramesacanthion provided by Smol, Muthumbi & Sharma (2014) specifically mentions its distinguishing characteristic is “spicules consisting of two portions, distal and proximal, articulating with each other”, while diverse nature of spicules in Mesacanthion is missing in its diagnosis. Even while knowing some spicules in Mesacanthion can be bipartite, it is imperative to put emphasis into the word “articulating” in diagnosis of Paramesacanthion species’ two portioned spicules. Most spicules of the genus Paramesacanthion clearly depict the two portions and its articulation from one another (i.e., Knee joint). While some species like P. marei, does not clearly show the articulation much like bipartite spicules found in Mesacanthion, they too can be distinguished as a Paramesacanthion species by its: 1. Outer labial and cephalic setae in front of anterior end of cephalic capsule and 2. Subcephalic setae at middle of cephalic capsule. Paramesacanthion species bear an extra ring of subcephalic setaes which can be confused with the true cephalic setae of Mesacanthion species. Wieser (1953) specifically mentioned this when erecting the genus, not to confuse “the subcephalic setae of the male with the true cephalic setae since the former occupy that level of the head on which in Mesacanthion the insertion of the cephalic setae takes place” (p.80). As such, the new species has a paired spicule which are symmetric from one another. They are bipartite with distal portion slightly shorter than proximal part. Distal and proximal is divided by a seam which seems to thicken or “arm” around the distal portion of the spicule. Proximal end arcuate and distal end set with pointing spine or a “barb”.

Table 2 Comparison of diagnostic morphological characters of all Mesacanthion species.

Species with spicules shorter than 2 anal body diameters marked with asterisk. Males only, morphometric values rounded.

Species	Body length [μm]	a	b	c	c’	Length of Setae	Spicule length [μm] (spicule length as arc/abd) left/right if applicable	Spicule type	Gubernaculum (length [μm])	Supplementary organ/papilla distance from cloacal opening [μm] (supplementary organ distance from cloacal opening/abd)	
						Inner labial Setae	Outer labial setae/cephalic setae					
Mesacanthion africanthiformeWarwick, 1970*	2370–4490	65.8–81.8	4.1–5.4	16–19.6	4.5–4.9 calc	6–8	24–41/10–20	20–33 (0.6–0.7 calc)	Symmetrical/unipartite	Present (10–13)	Absent	
Mesacanthion africanum Gerlach, 1957	3345	33	6.1	12.6	3.7	6.5	15	85/180 (1.2/2.5 calc)	Asymmetrical/bipartite/striated	Present (53/44)	Present (88)	
Mesacanthion agubernatusVitiello, 1971*	3120	34.6	3.7	21.5	3.2	8	14–19	41 (0.9)	Symmetrical/unipartite	Absent	Present (155)	
Mesacanthion alexandrinusNicholas, 1993	1450–2460	33–43	3.1–3.6	14–22	3.4–4.5	12–13	27–29/12–16	79–86 (2.5–3.6)	Asymmetrical/unipartite	Present (not measured)	Present (64–70)	
Mesacanthion arabiumWarwick, 1973	5780–6250	30.4–37.0	4.8–5.2	16.1–18.4	3.7–3.9 calc	23–25	56–65/27–32	570–610 (6.2–6.8 calc)	Unclear/unipartite/striated	Present (120–127)	Present (220–230)	
Mesacanthion arcuatileWieser, 1959	No male described or measured	
Mesacanthion armatumTimm, 1961*	1630–1940	23–51.1	4.5–5.7	43.2–51	1.5–2	5	14/9	41 (1.4 calc)	Symmetrical/unipartite	Present (not measured)	Absent	
Mesacanthion audaxDitlevsen, 1918; Filipjev, 1927*	3700	57	4.8	14.5	3.2	Not measured	Not measured	143 calc (1.8)	Symmetrical/bipartite	Present (not measured)	Present (178)	
Mesacanthion banaleFilipjev, 1927; Gerlach & Riemann, 1974	No male described or measured	
Mesacanthion brachycolleAllgén, 1959	No male described or measured	
Mesacanthion brevisetaFilipjev, 1927; Gerlach & Riemann, 1974*	3960	23	4	12	3 calc	10	20/15	165 (1.5)	Symmetrical/unipartite	Present (60)	Absent	
Mesacanthion caveiInglis, 1964	4200	38.2	4.2	17.5	3.75 calc	13	59/35	510 (8.0 calc)	Symmetrical/unipartite	Present (38)	Present (161)	
Mesacanthion ceeumInglis, 1964	3500	41.7	4.9	13.5	5.8 calc	Not measured/ mentioned	59	430 (9.0 calc)	Symmetrical/unipartite	Present (31)	Present (121)	
Mesacanthion conicumFilipjev, 1918; Filipjev, 1927	No male described or measured	
Mesacanthion cricetoidesWieser, 1959	No male described or measured	
Mesacanthion diplechmaSouthern, 1914; Filipjev, 1927	3330–3980	39.8–43.8	5.4–5.9	12.6–14.8	4.0–4.6 calc	11	45/35	95 (1.2 calc)/500–598 (7.8 calc)	Asymmetrical/bipartite/striated	Present (not measured)	Present (80)	
Mesacanthion ditlevseniFilipjev, 1927; Gerlach & Riemann, 1974*	3580–6250	31.2–38	4.3–5.7	14–17.9	3.4–4.8, calc	12–16	21–26/not measured	87–100 (1.4–1.8)	Symmetrical/bipartite	Present (43–47)	Present (155–172 calc)	
Mesacanthion fricumInglis, 1966*	1650	42.3	3.75	9.07 calc	5.05	13	96/51	40 (1.1 calc)	Symmetrical/unipartite	Small/”uncertain”	Present (78)	
Mesacanthion heterospiculumSergeeva, 1974	2394–2398	23.7–23.9	4.4–4.6	13.6–14.2	Not measured	6.2	Not measured	109/54 (2/1)	Asymmetrical/striated	Present (19)	Present (35)	
Mesacanthion hirsutumGerlach, 1953*	1155–1532	40–49	3.4–3.9	9–12	4.3–6.1 calc	8–14	22–24/14	21–33 (1.3)	Symmetrical/unipartite	Absent	Present (45–100)	
Mesacanthion infantileDitlevsen, 1930 De Coninck & Schuurmans Stekhoven, 1933*	3230–5400	23.7–24.8	4–4.7	14.0–19	2.5–3	10–15	36–54 calc/∼20–35	112 calc (abd not given or depicted)	Symmetrical/bipartite	Present (not measured)	Absent	
Mesacanthion karenseFilipjev, 1927; Gerlach & Riemann, 1974*	1750	35–39	4.5–4.9	12–16	4.1–6	10	33–36/24–26	24 (1)	Symmetrical/unipartite	Present (14)	Not described, not depicted	
Mesacanthion kerguelenseMawson, 1958*	3500–9000	20.5–40	3.5–5.7	25–26	1.3–1.5	8	40–50/25-30	150–200 (1.9 calc)	Symmetrical/unipartite(with tapering point)	Present (not measured)	Present (proximal end of spicule)	
Mesacanthion longispiculumGerlach, 1954	2228–2575	49–55	3–3.3	18–25.8	2.7–3.8	11–17	33–38/13–16	75–143 (3.0–4.0 calc)	Symmetrical/unipartite	Not described, not depicted	Present (87–90)	
Mesacanthion longissimesetosumWieser, 1953*	3260–4270	29.2–31.7	5.1–5.3	11.1–13.2	4.2–4.5	12	65–70/40	83 (1.1)	Symmetrical/unipartite	Present (39)	Present (166)	
Mesacanthion luciferFilipjev, 1927; Gerlach & Riemann, 1974*	4390	26–30	4.3–4.6	10.7	4.1 (calc)	10	22–23	155 (1.5)	Symmetrical/unipartite	Present (55)	Present (∼300)	
Mesacanthion majusFilipjev, 1927; Gerlach & Riemann, 1974*	2840–3170	26.0–33.9	4.2–4.9	11.7–12.8	4–4.3	11.5–12	40/26	80 (1.35)	Symmetrical/unipartite	Present (27)	Present (134)	
Mesacanthion marisalbiPlatonova & Galtsova, 1976*	2992–4037	45.4–52.4	4.3–5.5	16–20.1	4.4–4.6 (calc)	6–8	not measured/61.2–64.0	56.7 (1.4 calc)	Symmetrical/unipartite	Present (21.6)	Present (126.9)	
Mesacanthion monhysteraGerlach, 1967*	1833	48	3	9.6	6.3 calc	12–13	23–25/9–10	25 (0.8 calc)	Symmetrical/unipartite	Absent	Present (85)	
Mesacanthion obscurumGagarin & Klerman, 2006	2163–3148	19–32	3.5–5.2	12.4–18.1	3.0–4.2	7–10	37–43/23–27	70–81/269–310 (1.4/5.2 calc)	Asymmetrical/bipartite/striated	Present (28–35)	Present (36–59)	
Mesacanthion paliWieser, 1959*	2160	54	3.3	15.4	2.8	24	84/8	62 (1.3 calc)	Symmetrical/unipartite	Present (26)	Present (78)	
Mesacanthion pannosumWieser, 1959*	4100	41	6.3	13.7	3	11–12	24–25/22	135 (1.5 calc)	Symmetrical/unipartite	Present (66)	Present (180)	
Mesacanthion propinquumGagarin & Klerman, 2006	2076–2674	19–30	4.0–5.1	10.8–15.8	2.7–4.5	4.0–4.5	11–15/7.0–8.5	70–77/239–308 (1.2/4.2 calc)	Asymmetrical/bipartite	Present (26–30)	Present (28–31)	
Mesacanthion proximum Gerlach, 1957*	1340	54	3.1	9.6	7–8	11	20/7–10	20 (1.1)	Symmetrical/unipartite	Absent	Present (46)	
Mesacanthion rigens Gerlach, 1957*	1680	52	3.3	14.6	4	8	25/8	25 (0.9)	Symmetrical/unipartite	Absent	Present (70)	
Mesacanthion southerniWarwick, 1973	3280–3900	32.1–33.9	5.6–5.8	14.4–14.9	4.2–4.5	12	57/39–48	67–80/290–320 (1.2–1.5/5.3–5.6 calc)	Asymmetrical/bipartite/striated	Present (40–51)	Present (82–100)	
Mesacanthion studiosumInglis, 1964*	5500–5900	48.3–53.7	4.1–4.3	16.0–18.7	4.1–4.8	19–20	48–50/18–20	68–81 (0.9–1.0 calc)	Symmetrical/unipartite	Absent	Present (129–159)	
Mesacanthion tenuicaudatum (Ssaweljev, 1912) De Coninck & Schuurmans Stekhoven, 1933	6000	45–50	5	22	Not measured	Not measured	Not measured	45 (adb not given or depicted)	No depiction (”chitinized”)	”Unclear”	Present (∼22.5)	
Mesacanthion virile (Ditlevsen, 1930) De Coninck & Schuurmans Stekhoven, 1933*	4400	25	5	17	Not measured	16 calc	50–61 calc	163 calc (adb not given or depicted)	Symmetrical/unipartite	Present (not measured)	Present (232 calc)	
Mesacanthion jejuensissp. nov.*	2703–3723	35.6–46.6	4.5–5	11.7–12.9	4.0–5.7	11–15	43–59/18–34	72–85 (1.5)	Symmetrical/bipartite	Present (39–50)	Present (136–171)	

Metanemes are one character which was surprisingly not observed within the new species. While no species belonging to Mesacanthion have yet been described to date with description or depiction of metanemes, orthometaneme of dorsolateral kind was expected to be present within the new species prior to inspection. Diagnosis of the family Thoracostomopsidae (Filipjev, 1927) (according to Smol, Muthumbi & Sharma, 2014) specifically states “only dorsolateral orthometanemes with a robus scapulus but no caudal filament”. Species belonging to Mesacanthion’s most closely related genus, Paramesacanthion abyssorum Bussau 1995, was also recorded with presence of dorsolateral orthometanemes. Not only that, “coffee bean shaped epidermal glands” which were sighted alongside dorsolateral orthometanemes in P. abyssorum are very much present within the new species as well (Figs. 2A and 3A). Given that orthometanemes are subtler in their appearance compared to loxometanemes, it is quite possible that even other species of Mesacanthion already described, could have had them present. Despite it being more difficult to spot in older types due to their conditions, it’ll be important for future descriptions of any species belonging to the family Thoracostomopsidae to identify their metanemes.

After discovering the new species, the type locality was visited twice more in August and November of 2018, to obtain alcohol samples of the specimen for molecular analysis. While the efforts were unfortunately fruitless until now, we are hopeful that we will get the required specimen for additional molecular analysis in the future. It’ll be interesting to compare the relationship between close related species by the means of molecular phylogenetic data.

Conclusion

The discovery of Mesacanthion jejuensis sp. nov., has led to number of findings: (1) the new species closely related to M. pannosum, in terms of general morphology (bearing precloacal supplementary organ and gubernaculum) and having modified cervical setae flap. (2) the new species, like three other species within the genus (M. audax, M. ditlevseni, M. infantile), has a pair of bipartite spicule. (3) the diagnosis of the genus Mesacanthion has been updated to account for diverse nature of spicules. (4) the genus Mesacanthion has been reviewed and revised, transferring two species, M. brachycolle and M. ungulatum to species inquirenda, updating the total number of valid species to 39 species. While we were unable to obtain genetic data for the new species, further efforts will be made in order to investigate the phylogenetic relationship and placement of species within the genus Mesacanthion.

Supplemental Information

Table S1 Raw measurements of all specimens observed

All values rounded.

Click here for additional data file.

We thank Vadim Mokievsky (P.P. Shirshov Institute of Oceanology) and Jungho Hong for helping with the field work to collect samples on June 2018. We also thank Ana Carolina Vilas-Boas and one anonymous reviewer for their careful reading of our manuscript which greatly improved this manuscript.

Abbreviations

a body length/maximum body diameter

abd anal body diameter

b body length/pharynx length

c body length/tail length

calc calculated or measured from published measurements and/or figures

c’ tail length/anal body diameter

Additional Information and Declarations

Competing Interests

Author Contributions

Data Availability

New Species Registration

The authors declare there are no competing interests.

Raehyuk Jeong conceived and designed the experiments, performed the experiments, analyzed the data, contributed reagents/materials/analysis tools, prepared figures and/or tables, authored or reviewed drafts of the paper, approved the final draft.

Alexei V Tchesunov conceived and designed the experiments, analyzed the data, contributed reagents/materials/analysis tools, authored or reviewed drafts of the paper, approved the final draft.

Wonchoel Lee conceived and designed the experiments, contributed reagents/materials/analysis tools, authored or reviewed drafts of the paper, approved the final draft.

The following information was supplied regarding data availability:

The raw measurements are available in Table S1.

All specimens are deposited in National Institute of Biological Resources (South Korea). Holotype (NIBRIV00008488276), male paratype 1 dried specimen (NIBRIV00008488280), male paratype 2 (NIBRIV00008488279), male paratype 3 (NIBRIV00008488278), allotype (NIBRIV00008488277), female paratype 1 dried specimen (NIBRIV00008488281), female paratype 2 (NIBRIV00008488278).

The following information was supplied regarding the registration of a newly described species:

Publication LSID: urn:lsid:zoobank.org:pub:989DF431-166A-4534-9A37-9AC408194DE7

Mesacanthion jejuensis sp. nov. LSID: urn:lsid:zoobank.org:act:EE4EB2FC-59DA-48D3-9C10-C9E5646AF0D9

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
