# Peer review of "Bibliographic revision of Mesacanthion Filipjev, 1927 (Nematoda: Thoracostomopsidae) with description of a new species from Jeju Island, South Korea"

_PeerJ, doi:10.7717/peerj.8023_

## Round 0.1 · original submission · Major Revisions

I have heard back from two taxonomic experts, both of whom have offered helpful comments that will help improve your work. In particular, Reviewer 1 has several major criticisms that must be addressed, particularly regarding the revision of the genus Mesacanthion. Please consider the comments carefully, and I look forward to seeing a revised version of your work.

Reviewer 1 ·

Basic reporting

No comment

Experimental design

Methods are correct and can be replicate.

Validity of the findings

I think the species is a new one, but the revision (bibliographic) is not useful without a more complete Mesacanthion general key or a table with male morphological measurements separate from females. The differential diagnosis is not well done and require more investigation.

If a key is included part of tables information can be go to supplementary material.

The head drawing are not clear, some details as lips configuration, details of mandibles and cephalic capsule are not included. The drawings and photos of spicules and gubernaculum are not enough to discern the presence of a seam.

Additional comments

Dear authors, I think you have a new Mesacanthion species, but much work and investigation are needed to demonstrate it. I include a list of corrections and doubts (following lines) that I had during the review of your paper. I hope this help you to elaborate a improved paper. The Mesacanthion revision must be improved with a more complete key that include a higher number of species or all.

Title
This is not a revision of Mesacanthion! From Material and methods you do not mention any visit to museums collections where original specimens are observed or you asked to museum collections for observe original material. This is a kind of bibliographic revision, but including a key only for four species! So neither is a bibliographic revision of Mesacanthion. To do a real bibliographic revision you must to detail the history of each species plus all the keys that have been done and offer a new key of Mesacanthion to help to understanding of this complex group.
You do not include a key for all the species! And the complete table 2 is not useful from taxonomy point of view, because you include measurements of males and females together!

Line 330.
“Lips well developed edges or lips narrowed and distally pointy” This details you describe are not seen in the drawings Fig 2A and 3A and from your drawings, they seems to be different in male than in female. You must to add a face view (light microscope or electronic microscope, SEM photo) of male and female head so this structures can be observed easily, or give detailed draw of this structures. Your written description is not enough if you do not document it. Lips structures are very important in this group and can help you to differentiate your species. How many papillae carry the lips? Are similar on the three lips? Are the same in males than in females?

Line 331
The cephalic capsule is not included in your drawings.
You must add it to show that the cephalic setae are in the midlevel of it.
This is easily to see do with new specimens in glycerin dyed with Bengal rose or Blue Nile (or both) before mounting. Your drawings and SEM photos are not enough.

Line 332.
From SEM photo 5B it is obvious that cephalic capsule is not vaguely define, because the contours are seen (but this is not enough), so please add new figure with the details of cephalic capsule. Specially the position of cephalic setae and amphids in relation to the cephalic capsule. If you could measure the cephalic capsule you can draw it!

Line 335.
“Mandible consisting of two rods distancing from one another anteriorly joined by anterior rod.”
How are the lateral edges of the anterior rod of mandibles? The mandibles have pointed teeth usually in the anterior arched rod, this teeth can be pointed to the lumen or laterally. They are only seen in apical view or sometimes in lateral view. In your drawing 3A in lateral view they seem to be pointed to the lumen! So please look carefully and give and an enlarged drawing with this important detail that is not included in your description.

Line 338.
Amphideal flap, inverse triangular, just posterior to the lateral outer labial seta at level of posterior end of cephalic capsule (Fig. 5B)
Are you sure it is an amphid? In M. pannosum, Wieser 1959 there are similar structures, and they are interpreted as modifications of cervical setae. In your SEM photo is easily seen that this triangular flap is in the same place of the sub lateral first crown, cervical setae. Please discuss this!
If this structure is not an amphid, where is the amphid? Please add a better drawing of cephalic capsule and amphid position. The amphid in this genus, in general, it is found in sub lateral position, at the end of the cephalic capsule. Could be overlapped with triangular flap?

Line 338
What are the two holes observed in SEM photo 5B? In Fig 5B, between the two crowns of cervical setae there are two pores. You must explain and describe what they are. It is a beautiful photo!

Line 345.
I cannot see the seam. There are a constriction between capitulum and spicule that is usually found in Mesacanthion and other related genera. In figure 2B the seam is not draw. From photos 4A and 4B I see a massive gubernaculum covering the spicule and a constriction in the proximal end of the gubernaculum. This is the seam you mention? Please add an enlarged figure with the details of the seam you see. Comparing with the other three species with real seam, your species is very different!

Line 377.
Differential diagnosis: The new species is most similar to M. ditlevseni as they both share
striking resemblance in overall morphology, including the stout and strongly based inner labial setae, similarly shaped gubernaculum (which was thought to be peculiar at the time of description of M. ditlevseni), as well as their body ratio being within range from one another (a, b, and c’). Probably this is true, but there are other species that are related. In M. ditlevseni the seam on the spicule in well seen in the middle of the spicule and the gubernaculum is triangular but short, very different to yours. I think that another related species is M. pannosum Wieser, 1959, because it has pointed lips, cervical flaps, but no have seam. The presence or not of the seam probably is not a good diagnostic character because it appear in some species of Mesacanthion, in Fleuronema and in Paramesacanthion, three genera very different. I think the presence of flap cervical setae is important!


Line 397.
“Amphideal flap observed for the first time in the genus.”
This is not true, see M. pannosum, Wieser, 1959.


Line 410.
I think this key have no sense and is not useful.
I think is wrong to include your species in this group that probably is not a real group, because the seam appear in very different genera! To demonstrate the seam probably you need to take more photos to other male specimens and draw them properly. The seam you mention: could be the end of a massive gubernaculum that for diffraction deform the spicule?
My suggestion if you want to do a real bibliographic revision that would be useful to future taxonomy is to do a new males key including all species Mesacanthion (this is the best but complicate) or at least all those species of Mesacanthion having spicules smaller than 2 anal diameters (*), independently of having or not seam. I think they are around 21 species, but probably I am wrong you must to check.
(*) Despite this is a relative measure and can be discussed, from my experience in very old description species, it is more reliable than measurement in microns.

Line 420.
From my point of view your species is related to M. pannosum Wieser, 1959 and to other species with spicule length smaller than 2 anal diameters, also it is related to M. ditlevseni (Filipjev, 1927) Gerlach & Riemann, 1974. You must do a new discussion basing it on your new key.


Figures
Fig. 1
OK
Fig. 2
Add a face view of male head, add draw of mandibles, add draw of cephalic capsule and position of cephalic setae and amphids, and see comment in text.
Fig. 3
If you have material idem to male. But if not you can obviate this.
Fig. 4
Add a detailed figure with the seam or observe if you interpreted it correctly.
Fig. 5
Explain in text all the details observed in the SEM photo.
Fig. 6
Take out this figure it has no sense!
Table 1
OK
Table 2
“Males and females joined, morphometric values rounded”
This is terrible confusing and is not useful for taxonomy, you must to separate measurements of males and females. Female’s variability in fertile stage is great!
You can do it in one table or separate tables. You must add one or both tables in supplementary material, it has no sense to include in text if you add a good key.

·

Basic reporting

The authors present a relevant and meaningful study, however it is not yet clear the choice of the genus. I suggest describing in more detail the genus under revision both in the abstract as in the introduction. Specially in the abstract, the most important findings of the revision were not mentioned. A brief explanation about the changes of the systematics of Mesacanthion could enrich the introduction.
The English used throughout the article must be revised, since the comprehension of some phrases was not clear (e.g. “Diagnosis of the genus Mesacanthion provided by Smol et al. 2014, which provides sufficient description as is, was given a slight update based on our review and findings”).
The figures have sufficient resolution and appropriated descriptions and labels, with minor changes. There are references in-text citations with and without commas.

Experimental design

The methodology could be differently structured, dividing it in three subsections (e.g. 1 – field surveys and morphological study; 2 – systematic revision; and 3 – nomenclatural acts). The methodology used for the revision of Mesacanthion must be detailed.

Validity of the findings

I suggest to better structure and to cite references in the diagnosis and in the discussion of the diagnosis of Mesacanthion. A special attention must be dedicated when considering invalidate any species, mainly when the type material was not consulted. The arguments must be very convincing in making this decision. I understand that the taxonomic validity of M. ungulatum needs a further investigation.

Additional comments

Other details are specified in the PDF document.

---

## Round 0.2 · accepted · Accept

Two taxonomic experts have re-reviewed your paper and found it to be well edited. I agree, and am pleased to move this into production.

Reviewer 1 ·

Basic reporting

The manuscript has been corrected. Literature provided and added is correct.
Only two simple grammar mistakes and one key modification need to be corrected. Please see manuscript attached.

Experimental design

I am sure that now the new species is well described.
The changes added in figures, photos and in descriptions are good now.
I agree with authors decisions in particular in key construction. But please correct lines 606-607 in the Mesacanthion key, you must to inform that you are speaking of precloacal sub ventral setae. Please see manuscript attached.

Validity of the findings

I am sure that now the new species is well described.

Additional comments

Dear authors your paper now is complete and can be published because your new species is well characterized. I recognize your effort to improve the manuscript. Please correct the three mistakes noted in the manuscript attached. From my point of view the paper must be accepted.

Annotated reviews are not available for download in order to protect the identity of reviewers who chose to remain anonymous.